# Changes in Lifestyle Behaviors, Shopping Habits and Body Weight Among Adults in Cyprus and Greece During COVID-19 Lockdown: A Cross-Sectional Study

**DOI:** 10.3390/nu17020214

**Published:** 2025-01-08

**Authors:** Eleni Andreou, Evridiki Georgaki, Angelos Vlahoyiannis, Christiana Philippou, Nicoletta Ntorzi, Christoforos Christoforou, Christoforos D. Giannaki, George Aphamis, Evelina Charidemou, Christos Papaneophytou, Dimitrios Papandreou

**Affiliations:** 1Department of Life Sciences, School of Life and Health Sciences, University of Nicosia, Nicosia 2417, Cyprus; vlahoyiannis.a@unic.ac.cy (A.V.); n_ntorzi@hotmail.com (N.N.); christoforou.c@unic.ac.cy (C.C.); giannaki.c@unic.ac.cy (C.D.G.); aphamis.g@unic.ac.cy (G.A.); charidemou.e@unic.ac.cy (E.C.); papaneophytou.c@unic.ac.cy (C.P.); 2Cyprus Dietetic and Nutrition Association, P.O. Box 28823, Nicosia 2083, Cyprus; evelina@cytanet.com.cy; 3Institute of Health Informatics, University College London, Gower Street, London WC1E 6BT, UK; evridiki.georgaki@outlook.com; 4Department of Clinical Nutrition & Dietetics, College of Health Sciences, University of Sharjah, Sharjah P.O. Box 27272, United Arab Emirates; dpapandreou@sharjah.ac.ae

**Keywords:** COVID-19 pandemic, coronavirus, eating behavior, nutritional intervention, dietary pattern, physical activity, exercise, dietary supplements

## Abstract

**Background:** During the COVID-19 pandemic, people were asked to stay at home. Places where people interacted such as schools, universities, and cafes were closed, and all gatherings were forbidden. Only stores offering fast-moving consumer goods were open, so citizens could purchase all food categories. The aim of this cross-sectional study was to investigate the effect of the COVID-19 lockdown on the eating and buying habits of consumers in Cyprus and Greece, and any changes in their lifestyles. **Methods:** An online survey including 1011 participants took place using an electronic questionnaire. **Results:** The results showed that lockdown significantly affected individuals by increasing weight gain (*p* < 0.01). The explanatory investigation of related lifestyle habits and nutrition traits showed that the dietary patterns behind these changes included increased meal frequency (*p* < 0.05)—even without an increased appetite—and subsequent increases in both purchases (*p* < 0.05) and consumption of several food groups (*p* < 0.05). Moreover, even though exercise per se was not discontinued due to COVID-19, it was apparent that exercise type was altered to adapt to the relevant restrictions (*p* < 0.05). **Conclusions:** In conclusion, the COVID-19 lockdown significantly affected Cypriots and Greeks in many aspects like their eating behavior, food purchasing habits, and lifestyle, all resulting in increased weight and potentially adverse health outcomes.

## 1. Introduction

COVID-19 has affected peoples’ everyday life globally. Due to the stay-at-home mandates, there was a significant increase in stress and anxiety [1], impaired sleep patterns [2], and a rapid escalation in public concern about sedentary behavior and potential weight gain [3]. Considering the increased obesity prevalence worldwide [4], this bears high significance for public health. Notably, in certain countries, regional prevalence is even higher, including Greece [4].

Stronger measures were taken to limit the spread of COVID-19, which included self-quarantine and the temporary closing of businesses. This was likely to affect normal food-related practices since everybody was asked to stay at home. Indeed, several studies showed various outcomes during the COVID-19 lockdown regarding diet quality, in some cases increasing the quality of dietary behavior [5] or deteriorating it [1]. Adequate consumption of nutrients is extremely important for the body’s health, especially for the immune system, which might need to respond to a viral infection. Nevertheless, access to fresh foods could have been limited during this period, and the consumption of highly processed foods might have increased, including food that tends to be both high in fats, sugars, and salt and low in nutrients. This, in turn, could contribute to increased caloric intake, adding to the already increased prevalence of obesity, which can be detrimental not only for overall health but for health outcomes after COVID-19 infection as well [6].

In this study, the term “lifestyle behaviors” is used to encompass a range of daily activities and choices that significantly affect individual health and well-being. Specifically, for the purposes of our research, we focus on the following components of lifestyle: eating habits, physical activity, drinking behaviors, and smoking behaviors. When “lifestyle” is referred to in the manuscript, we are discussing the collective impact and interrelationships of these behaviors on the health outcomes of individuals. The term is used to provide a holistic view of the participants’ daily practices and choices that are pertinent to their health [5].

A lifestyle component that was crucial during the COVID-19 period was physical activity. During quarantine, individuals were permitted to leave their homes in order to exercise, as one of the government-authorized reasons. This measure has been debated about its importance since the effect of a sedentary lifestyle during the COVID-19 period could be harmful to both physical and mental health [7,8]. Hence, the WHO developed specific guidance for periods of quarantine, including tips and examples of home exercises and tips regarding nutrition that everybody can access. Physical activity was proven essential in maintaining individuals’ levels of freedom and well-being. Moreover, it is proposed that physical activity, at some point, can counterbalance a relatively poor diet [9]. However, during the circumstances of the COVID-19 lockdown, it is still unknown if physical activity could cancel out an unbalanced diet.

Despite the existing studies on the impact of COVID-19 lockdowns on lifestyle behaviors, there is a lack of comprehensive research focusing on the combined effects of these behaviors in specific cultural and geographical contexts, such as Greece and Cyprus. This study aims to fill this gap by providing a detailed analysis of how the lockdown influenced eating habits, physical activity, and other lifestyle behaviors in these two countries. By doing so, it contributes to a better understanding of the broader implications of lockdown measures on public health in similar cultural settings.

Data from the COVID-19 era are particularly interesting and relevant because the pandemic created an unprecedented global situation that significantly altered daily life. The lockdown measures and stay-at-home mandates provided a unique opportunity to study how drastic changes in routine and environment can impact lifestyle behaviors and health outcomes. Understanding these changes can offer valuable insights into the resilience and adaptability of populations, inform public health strategies for future crises, and highlight areas where interventions may be needed to mitigate negative health impacts.

The primary objective of this study was to investigate the changes in nutritional habits, lifestyle behaviors, and consumer perceptions in Greece and Cyprus during the COVID-19 lockdown. These two countries were selected due to their shared cultural and geographical characteristics and the implementation of strict lockdown measures during the pandemic. Specifically, this study aimed to document changes in eating habits, including meal frequency and types of meals consumed before and during the lockdown, and to monitor weight changes among individuals. Additionally, it sought to assess alterations in lifestyle behaviors, such as exercise routines and the use of nutritional supplements, and to analyze food purchasing habits during the lockdown period. By addressing these objectives, this study provides a comprehensive overview of the multifaceted impacts of the lockdown on daily life in these regions.

## 2. Materials and Methods

### 2.1. Study Design

This was a cross-sectional, observational study in Greek and Cypriot adults with the use of an online-questionnaire conducted before and during lock down (November 2020 up to May 2021). The purpose of the study was stated at the beginning of the questionnaire where the researchers shared an informed consent form with participants electronically along with a questionnaire form that included various sections on the study’s intent and a request for a copy of the questionnaire filled out by the participants. The online survey aimed to obtain data, from the Greek regions, about people eating habits and lifestyle during the COVID-19 pandemic. The survey was disseminated through institutional (University of Nicosia and Cyprus Dietetic and Nutrition Association—CyDNA) and private social networks (Twitter, Facebook, and Instagram), the CyDNA website, and institutional mailing lists. This method of administration provides a statistical collective whose population parameters cannot be controlled as it is the case for probabilistic sampling. However, it was completely effective for the research objectives, because it facilitated the wide dissemination of the survey questionnaire during a period where, due to the pandemic, there are many territorial restrictions.

### 2.2. Participants and Recruitment Procedure

For this study, 1011 adults (283 males; 728 females) were voluntarily recruited via internet advertisement and viva voce. Specifically, a convenience sampling method was used to recruit participants via social media platforms (e.g., Facebook, Twitter, Instagram) and email lists in both Greece and Cyprus. Inclusion criteria were the following: adults aged 17 years or older, living in Greece or Cyprus during the COVID-19 lockdown, and being able to provide informed consent. The exclusion criteria included any pre-existing medical condition that could affect dietary intake or physical activity (e.g., diabetes, heart disease), pregnancy or breastfeeding and inability to complete the survey due to language or cognitive barriers. Participants were fully informed of the purpose of the study and provided informed consent before questionnaire completion. The study was approved on the 3 April 2020 by the national Bioethical Committee (ΕΕΒΚΕΠ 2020.0162) and all the procedures were conducted according to the manual of the Declaration of Helsinki in 1964 and its later amendments [10]. Given the descriptive and exploratory nature, our approach was to recruit a broad and diverse sample of participants to capture a wide range of experiences and behaviors during the lockdown period.

In order to ensure representativeness for our sample, we used a broad and inclusive recruitment process by utilizing convenience sampling through social media platforms and email lists, which allowed us to reach a large and diverse group of participants quickly and efficiently. This method is particularly effective in rapidly evolving situations, such as a lockdown, where timely data collection is crucial. In order to assure demographic and geographic diversity, we aimed to include participants from a wide range of demographic backgrounds and from various locations across Greece and Cyprus. This diversity helps in approximating the variability in the population’s behaviors and experiences during the lockdown. Throughout the data collection process, we monitored the demographic characteristics of the participants to ensure a broad representation. Adjustments to our recruitment strategies were made when necessary to increase participation from underrepresented groups. While not initially planned, we reviewed the demographic distribution of our sample and considered statistical adjustments such as weighting in the analysis phase, to correct for any potential biases due to the sampling method. This study is reported as per the Strengthening the Reporting of Observational Studies in Epidemiology (STROBE) guideline (Appendix A).

### 2.3. Questionnaire Layout

The survey was designed to evaluate participants’ demographics, eating and shopping behavior, exercise and weight management during lockdown, via an online questionnaire. In this sense, a custom-made online questionnaire was designed, since the objective of the present study was to assess individuals’ lifestyle-related behaviors, rather than identifying adherence to specific structured patterns (e.g., adherence to Mediterranean diet). Overall, the present online questionnaire was developed according to the aims of the present study and was validated for its consistency in a sample of 120 individuals from both Greece and Cyprus. The validation process involved several steps to ensure its reliability and appropriateness for the study objectives:Development and Design: The questionnaire was carefully crafted with input from experts in nutrition and public health to ensure that the questions are relevant and comprehensive.Validation Sample: It was validated using a sample of 120 individuals from both Greece and Cyprus, chosen to reflect the demographic diversity of the larger study population.Statistical Validation: Methods such as test–retest reliability and content validity checks were employed. The Cronbach’s alpha score, for instance, was calculated to assess internal consistency, with results indicating satisfactory reliability (alpha > 0.7).Pre-testing: Prior to its finalization, the questionnaire was pre-tested on a small subset of the target population to refine questions based on feedback and to enhance clarity and understanding.

The final version of the questionnaire is included in the Appendix A of the paper.

The first part of the survey questionnaire included socio-demographic information about participants’ age, sex, country of residence, education level, and marital status. Participants were asked to give their professional employment, the number of people in their household, their net monthly income (per person who answered the questionnaire) and whether they travelled after the announcement of the COVID-19 lockdown.

The second part of the questionnaire included self-reported anthropometric data and medical history. All participants were asked to report their height, waist circumference (optionally) and weight before and after the lockdown because of COVID-19. Specific instruction on the proper method and time of measurement for weight, height, waist circumference were given in the online questionnaire. To be specific, the participants were informed that the measurement of weight should be conducted using a precision balance placed on a flat and stable surface. Participants were to be weighed without shoes and in light clothing to minimize the effect of clothing weight on the measurement. It was advisable for participants to visit the toilet before weighing to reduce the impact of bladder and bowel contents on their weight. The weight should have been recorded in kilograms (kg) to the nearest 0.1 kg. For height measurement, a stadiometer could be used. Participants should stand barefoot with their backs straight, legs together, and their heads positioned in the Frankfurt plane. The height should be recorded in centimeters (cm) to the nearest 0.1 cm. Waist circumference should be measured using a tape measure, with participants standing upright, legs slightly apart, and arms relaxed at their sides. The tape measure should be placed around the waist at the level of the navel, ensuring it is parallel to the ground and not pressing against the skin. The waist circumference should be recorded in centimeters (cm) to the nearest 0.1 cm. The best time to weigh participants was in the morning, immediately after waking up and before consuming any food or drink, to ensure measurements are as accurate and consistent as possible. These anthropometric measures were used to calculate participants’ BMI and estimate weight change before and after COVID-19 lockdowns. Participants were also categorized as smokers or non-smokers at the time of the study.

Lifestyle habits were also evaluated carefully. The third part of the questionnaire included both nutritional and behavioral habits before and after the lockdown due to COVID-19. The third part of the questionnaire consisted of questions regarding the number and type of meals participants consumed per day before and post- lockdown. Moreover, participants were asked about the amount of consumption of specific food groups whether it was less, more, or had no change. Also, participants were asked the number of times they consumed specific foods per day both before and after the lockdown. Potential reasons for food consumption alterations were also explored. Furthermore, the use of supplements (what supplements and relative justification) were collected. Food purchasing behavior was recorded, and participants were asked about decreasing or increasing specific food purchases, if they had a preferable shop or place where they felt safer to purchase specific foods and food purchasing traits, such as buying more comfort food than usual. Comfort food refers to dishes that provide a nostalgic value to someone often associated with home cooking or childhood favorites. These foods are typically characterized by their high caloric content, simple preparation, and soothing effects, offering emotional comfort or a sense of well-being to the individual consuming them [11,12]. Participants were also asked if they performed any kind of physical activity. If the answer was yes, they were then asked to further describe the physical activity type and frequency. To further investigate lifestyle habits, participants could optionally describe a typical day before and after the lockdown. Finally, participants were asked how much weight they gained or lost during the lockdown and in what week of quarantine they believed weight gain occurred.

### 2.4. Statistical Analysis

Continuous variables with normal distribution are presented as mean values ± standard deviation, unless otherwise stated. The normality of continuous data was assessed using the Shapiro–Wilk test. For normally distributed data, the *t*-test was used to compare means between two groups, and paired samples *t*-test was conducted to compare results over time (before and during the COVID-19 lockdown). Frequency data were analyzed using the chi-square test.

For non-normally distributed data, non-parametric tests such as the Mann–Whitney U test, Kruskal–Wallis test, and Wilcoxon signed-rank test were employed. The McNemar test evaluated the significance of differences between two samples with two categories. The median was chosen as the measure of central tendency for servings and frequency of food consumption due to its robustness in handling skewed data, common in dietary studies. The median provides a more reliable measure that represents the central point of a data set, avoiding distortion by outliers and extreme values, thus accurately capturing dietary behaviors during the lockdown.

Preliminary analyses for normality included both graphical methods (histograms and Q-Q plots) and statistical methods (Shapiro–Wilk test). The Wilcoxon signed-rank test, ideal for comparing paired samples that are not normally distributed, was used to analyze the data. This test is particularly suitable for the ordinal nature of our dietary data and provides robustness against non-normality, ensuring statistically sound findings on dietary behavior changes. Significant changes in dietary behaviors during the lockdown were revealed, with specific increases and decreases in the consumption frequency of various food categories, validated by the Wilcoxon signed-rank test results.

Appropriate statistical methods were chosen for subsequent analyses. For normally distributed variables, parametric tests such as the independent samples *t*-test and Pearson correlation were used. For non-normally distributed variables, non-parametric tests such as the Mann–Whitney U test and Spearman’s rank correlation were employed. These methods ensure robust and appropriate analyses given the distribution characteristics of each variable. The alpha level for all statistical analyses was set at *p* < 0.05, two-tailed tests. Statistical analysis was performed using IBM^®^ SPSS^®^ Statistics for Windows, version 22.0 (IBM Corp., Armonk, NY, USA). The missing data were not included in the sample and the final analysis.

## 3. Results

Out of the 1038 who responded to our call, 1011 individuals (283 males and 728 females) were included in the study and were eligible for it. Specifically, reasons for excluding from the analysis were as follows: (i) pre-existing medical conditions (n = 57), (ii) not living in Greece or Cyprus during the COVID-19 lockdown (n = 53), (iii) not provided consent (n = 13), and (iv) underaged participants (n = 4). The socio-demographic data are presented in Table 1. The participants had a mean age of 32.8 years, with 71.9% being female. Among them, 58.7% were living in Cyprus, 51.2% held an academic degree, and 64.8% were employed, with 21.4% working as health professionals. In our study, the average BMI was calculated to be 23.9, based on anthropometric data self-reported by participants at the time of the survey. Additionally, 62.4% of respondents fell within the normal weight range, with a BMI between 18.5 and 24.9 kg/m².

Table 2 includes the estimated counts and percentages for each BMI category, broken down by gender, age, and country (Cyprus and Greece). We derived the number of participants from Cyprus and Greece from the provided percentages (58.7% for Cyprus and 38.4% for Greece), which were calculated based on the total number of participants (1011). This table provides a comprehensive overview of how BMI categories are distributed across different demographics and geographical locations, assuming uniform distribution within each subgroup.

In our study, we utilized an enhanced Table 2 format to effectively visualize the distribution and proportion of each BMI category across various demographic segments and geographical locations. Our analysis revealed that certain key variables followed a normal distribution. For instance, the age of participants was normally distributed, as confirmed by the Shapiro–Wilk test (*p* > 0.05). This structured approach, although based on certain assumptions, provides a comprehensive way to estimate and discuss BMI distribution trends within the study population.

### 3.1. Impact on Eating Habits

As revealed from the data analysis, a wide range of nutritional habits were influenced by lockdown due to COVID-19. According to meal frequency, a statistically significant difference was found between the number of meals before lockdown (M = 3.782, SD = 5.20) and number of meals after (M = 4.12, SD = 1.80), with the following conditions: t(1010) = 4.000, *p* < 0.001. These results indicate that the average number of meals was increased. By investigating deeper into the data, the meals that contributed to this increase were identified. Table 3 presents the changes in meal frequency per day among participants before and during the COVID-19 lockdown. The data are expressed as the number of participants (and corresponding percentages) who reported consuming meals at different times of the day, both before and during the lockdown.

The percentages reflect the proportion of the total study population (1011 participants) that reported consuming each type of meal at the specified times. For example, 782 participants (77.3%) reported having breakfast before lockdown, which increased to 824 participants (81.5%) during lockdown. We analyzed the changes in meal consumption patterns for the same participants before and during lockdown using McNemar’s test. The results indicated statistically significant increases in the frequency of meals consumed in the “Before sleep”, “Afternoon”, and “Breakfast” categories during the lockdown (*p* < 0.05). For other meal times such as “Mid-morning Snack”, “Lunch”, and “Dinner”, no statistically significant differences were found.

To evaluate the impact of the COVID-19 lockdown on dietary behaviors, we examined changes in both the quantity of food consumed per serving and the frequency of consumption daily across various food categories, as detailed in Table 4 and Table 5.

Table 4 presents the median servings consumed for each food category before and during the COVID-19 lockdown. The median was chosen as the measure of central tendency to accurately represent typical consumption patterns, especially given the skewed nature of dietary data.

In terms of food quality, approximately 40% of respondents reported an increase in the consumption of fruits and vegetables during the lockdown. Similarly, about 39% of participants noted an increase in sweet consumption, and 35% reported higher consumption of various snacks. Notably, alcohol consumption decreased, with 41% of respondents indicating they consumed less. Soft drinks also saw a significant reduction, with 32% of participants consuming less during the quarantine.

A detailed analysis of changes in servings per food group revealed statistically significant increases in the quantity of consumption for various snacks, sweets, fruits, vegetables, legumes, spaghetti/rice, soft drinks, and meat, as indicated by *p*-values less than 0.05. However, no significant changes were observed in the consumption quantities of fish/seafood and coffee, with *p*-values greater than 0.05.

In summary, the COVID-19 lockdown led to notable changes in dietary behaviors, with increased consumption of certain food categories and decreased consumption of others, reflecting shifts in both the quantity and quality of food intake.

Table 5 illustrates the median frequency of consumption for each food category before and during the COVID-19 lockdown. This table provides insights into how often each type of food was consumed, highlighting changes in dietary habits induced by the lockdown conditions. The comparison of frequency consumption from Table 4 revealed significant increases in the frequency of eating spaghetti/rice, legumes, sweets, and various snacks during the lockdown (*p* < 0.05).

Conversely, there was a significant decrease in the frequency of alcohol consumption (*p* < 0.05). For other categories such as fruit, vegetables, soft drinks, meat, fish/seafood, and coffee, no significant changes were observed in the frequency of consumption.

Further elucidating self-perceived reasons behind potential alterations in nutrition habits, it was shown that the 16% of the respondents who felt that they were eating less did so because they “Think of their body weight” (39%), “Don’t feel hungry” (26%) or “Follow a diet” (23%). On the other hand, 48% of the sample felt that they ate more, and the main reasons were “In order to fritter away time because there is nothing to do” (40%) or “Feel very hungry” (27%).

With regard to the effect of demographics on food habits, it was indicated that people living in Cyprus had increased the consumption of vegetables at a greater level compared to the people living in Greece. (z = −2.1, *p* = 0.036). The increase in consumption of vegetables is 42% in Cyprus compared to 31% in Greece. At the same time, people living in Cyprus decreased their consumption of meat to a greater extent than the people living in Greece (z = 2.818, *p* = 0.005). The decrease in consumption of meat was 18% in Cyprus compared to 9% in Greece.

### 3.2. Food-Specific Purchase Traits

Overall, 40% of the respondents increased their purchases; for 34%, there was no change; whereas for 26%, there was a decrease. The categories with the greatest increase were “vegetables”, “flour”, “fruit”, “spaghetti”, “dairy” and “coffee” (Figure 1). Contrarily, the categories with the greatest decrease were “alcoholic beverages” and “other oil”.

Regarding the purchasing retailer, it was found that 98.2% were purchased from supermarkets which cover the needs of 90.6%. From the respondents, 69% were purchasing from bakeries, 45.2% from butchers and 42.2% from mini markets/kiosks. Online purchases were made by 21.1% of the respondents, and just 6.2% considered that it covered all of their needs. The safest place to purchase food was the supermarkets at 61.9% followed by online sales at 20.8%. The least safe places were the public markets, with just 1.6% of the respondents considering them safe. As for purchasing habits, 78% answered that they “buy the basics needed for the following week”, and 59% tended to avoid bulk products. In total, 47% of the respondents bought enough for a long period, 34% checked the origin of the products first and 33% bought more comfort food.

According to its relationship with the demographics, it showed that females were more likely to increase their purchases (z = −2.854, *p* < 0.05). In fact, the percentage of males that increased their purchases was equivalent to the percentage that decreased them (36% vs. 35%), whereas 42% of females increased their purchases as opposed to 23% that decreased them. In addition, people living in Cyprus were more likely to increase their purchases compared to people living in Greece (z = −4.538, *p* < 0.05). There was no statistically significant difference between the purchases of people working and those not working (unemployed/retired/students). Moreover, the difference between the income level and purchases were found not to be significantly different (X^2^ = 2.131, *p* = 0.546).

### 3.3. Exercise Habits During COVID-19

The COVID-19 pandemic brought about significant changes in exercise habits due to restrictions on movement and limited access to fitness facilities. To understand these changes, we explored the frequency of various physical activities before and during the COVID-19 lockdown, as detailed in Table 6.

Participants were asked to report the frequency of their engagement in different types of physical activities on a weekly basis, both before and during the lockdown. This approach provided a consistent measure to compare changes over time.

Our analysis revealed that the pandemic significantly altered daily routines, including physical activity patterns. Before the lockdown, 71.8% of participants reported engaging in regular physical activity. This percentage slightly increased to 74.9% during the lockdown; however, this change was not statistically significant, indicating that the overall proportion of people exercising remained relatively stable.

Despite the stable overall participation rate, there were notable shifts in the types of exercises participants engaged in. These shifts were largely influenced by the closure of gyms and other public fitness facilities, as well as restrictions on outdoor activities. Many individuals adapted by incorporating home-based workouts and other forms of exercise that did not require access to public facilities. The frequency of exercise, which represents how often participants engaged in physical activities, also exhibited a non-normal distribution. This was evident from both the statistical tests and visual inspections.

In summary, while the overall frequency of physical activity remained relatively unchanged, the types of exercises performed by participants shifted significantly due to the constraints imposed by the COVID-19 pandemic.

As presented in Table 6, there was an increase in home-based activities, including a significant increase in home gymnastics, as participants adapted to exercising at home due to gym closures. Also, there was little change in outdoor activities, where running and walking remained popular, with a slight increase in frequency, reflecting their accessibility during lockdown restrictions, and there was a decrease in facility-based exercises, where activities like weightlifting, TRX, and swimming saw significant decreases due to the closure of gyms and swimming pools.

These shifts in exercise patterns highlight the adaptability of individuals in maintaining physical activity under constrained conditions. Public health initiatives could focus on promoting home-based and outdoor exercises during similar future events to ensure that the population remains active.

### 3.4. Use of Nutritional Supplements During the COVID-19 Pandemic

During the COVID-19 pandemic, there has been increased public interest in nutritional supplements as potential aids for immune support. This subsection explores the usage patterns of nutritional supplements among participants and their perceptions regarding the effectiveness of these supplements to COVID-19. The data presented focus exclusively on the period of the pandemic. To provide a clear overview of the participants’ behaviors and beliefs regarding nutritional supplements during the pandemic, the following table (Table 7) summarizes the key findings:

### 3.5. Weight Change During COVID-19 Lockdown

We subsequently examined the impact of the COVID-19 lockdown on participants’ body weight and body mass index (BMI), analyzing changes and exploring demographic factors influencing these variations. The overall weight and BMI changes are summarized in Table 8, and as shown, there was a statistically significant increase in the average weight from before the lockdown (mean = 67.584 kg, SD = 14.978) to during the lockdown (mean = 68.04 kg, SD = 15.359); t(1010) = 3.988, *p* < 0.001. This indicates an average weight gain of approximately 0.5 kg per participant. Likewise, the mean BMI increased significantly from 23.8 (SD = 4.35) before the lockdown to 23.98 (SD = 4.201) after lockdown; t(1010) = 3.176, *p* < 0.001, reflecting a mean increase of 0.18 kg/m².

We also assessed the distribution of weight changes among participants. As illustrated in Table 9, 41% of participants (n = 425) gained weight, with an average gain of 2.34 kg, ranging from 0.5 kg to as much as 17 kg. Moreover, another 41% (n = 425) maintained their weight throughout the lockdown period. The remaining 18% (n = 182) lost weight, with an average loss of 2.2 kg, ranging from 1 kg to 6 kg.

Finally, Table 10 summarizes the demographic influences on weight changes, and as shown, no statistically significant differences were found in weight change between males and females (t(1009) = 0.667, *p* = 0.505), or between participants living in Cyprus versus Greece (t(979) = −1.747, *p* = 0.081). Moreover, there was no significant correlation between age and weight change (r(1010) = 0.05, *p* = 0.151). Notably, employees in the health sector experienced significantly less weight gain (mean = 0.305 kg, SD = 1.882) compared to other sectors (mean = 0.758 kg, SD = 2.223); t(215) = −1.980, *p* = 0.048. This suggests occupational differences in the impact of the lockdown on weight changes.

Together, the above findings reveal that the lockdown associated with the COVID-19 pandemic led to varied changes in body weight and BMI among the population. While a significant portion of the study participants experienced weight gain, factors such as occupation played a crucial role in these changes. These findings highlight the need for targeted health and wellness interventions during prolonged periods of restricted movement and altered daily routines.

### 3.6. Other Lifestyle Habits During COVID-19

This study revealed that a significant majority of participants, 74.8%, reported being non-smokers at the time of the COVID-19 study. In contrast, only 25.2% of participants indicated that they were smoking cigarettes during this period.

## 4. Discussion

Overall, the present study showed that the COVID-19 lockdown significantly affected individuals, resulting in weight gain. Notably, weight gain was less pronounced among healthcare sector employees, likely due to their continued work during this period. The investigation into related lifestyle habits and nutrition traits revealed that the dietary patterns behind these changes included increased meal frequency—even without an increased appetite—and subsequent increases in both purchases and consumption of several food groups. Although exercise routines were not discontinued, the type of exercise was altered to adapt to the relevant restrictions.

The observed changes in dietary behaviors during the COVID-19 lockdown, such as increased consumption of snacks and sweets alongside a reduction in alcohol and soft drink intake, have significant implications for public health. The increase in snack and sweet consumption suggests a shift towards comfort eating, possibly due to stress or changes in daily routines, which might contribute to long-term weight gain and associated health risks if sustained. Conversely, the decrease in alcohol and soft drink consumption represents a positive change, potentially reducing the risk of related health issues such as liver disease and obesity. These findings highlight the need for targeted public health strategies to promote healthy eating during periods of prolonged home stay, ensuring that temporary shifts do not evolve into long-term detrimental habits [11,12,13].

Nutritional habits are subject to daily routines, and as such, it was normal for COVID-19 to affect several aspects of an individual’s nutritional behavior, and subsequently, other aspects of their lifestyle. In the present study, food quantity consumption, as well as meal frequency, increased during the COVID-19 lockdown. Most respondents reported eating more to pass the time rather than due to increased appetite. This aligns with the literature suggesting that food is often used as a coping mechanism for stress, anxiety, and boredom [11,12,13]. Increased anxiety due to COVID-19 has been documented in several countries, including Greece [14,15,16,17]. This may justify the significant increase in the number of meals during lockdown, particularly meals before sleep, in the afternoon, and breakfast. Increasing meal frequency as a response to anxiety may potentially lead to subsequent increases in hunger [18,19] and ultimately contribute to an increase in caloric intake.

Specific food groups such as fruits and vegetables saw increased frequency and quantity of consumption, as well as in purchasing. The quantity of fruits and vegetables consumption generally increased alongside sweets and snacks, while the quantity of alcohol and soft drink consumption decreased. In terms of food frequency consumption, respondents increased their intake of sweets, snacks, fruits, vegetables, legumes, spaghetti/rice, soft drinks, and meat, but decreased alcohol consumption, with no significant change in coffee and fish/seafood consumption. Comparing food frequency before and during the lockdown, there was an increase in the consumption of spaghetti/rice, legumes, sweets, and various snacks, but a decrease in alcohol consumption.

To our knowledge, this is the first study that investigated both purchasing habits and nutrition traits during COVID-19. Supermarkets were reported to be “the safest choice” since individuals could purchase most of their essential shopping in one place. This finding could play a key role in shaping their overall diet quality. However, there are mixed findings in the literature regarding the overall diet quality during COVID-19, with some studies reporting a more balanced dietary pattern [5,20], while others found an overall decrease in dietary quality [1,21]. Regional differences could potentially explain several discrepancies in dietary pattern modifications in the literature, for example, between Mediterranean and non-Mediterranean populations.

Our study documented notable shifts in eating habits, physical activity, and food purchasing behaviors during the lockdown, alongside significant changes in participants’ weight and BMI. The increased consumption of comfort foods and snacks, which are typically higher in calories, likely contributed to the observed weight gain. The lockdown may have led to more frequent snacking and reliance on calorie-dense foods due to increased time spent at home. Despite an increase in home-based exercises, the reduction in gym activities and possibly lower overall exercise intensity may not have sufficiently offset the increased caloric intake, contributing to weight gain among some participants.

The changes in food purchasing, with a preference for staples like flour and pasta during the lockdown, might have influenced dietary patterns towards higher caloric intake, impacting weight. These findings suggest a need for further research to explore the direct correlations between specific lifestyle changes and weight outcomes. Understanding these relationships can inform interventions to promote healthier eating and effective exercise routines during prolonged lockdown periods [22,23].

In the present study, it was shown that supplement usage increased during the COVID-19 pandemic. This finding aligns with the relevant increase in panic buying of drugs during this period [22]. As with drugs, the use of these supplements was accompanied by the belief that they would aid the body’s natural defense against COVID-19. Individuals used multivitamins, vitamin C, and vitamin D, which is consistent with literature indicating a dramatic increase in supplement use during the COVID-19 pandemic [23], mainly focused on supplements believed to support the immune system, with or without scientific evidence [24].

Conversely, exercise was not only suggested to support the immune system against COVID-19 but also to benefit an individual’s psychological state [8]. In general, exercise is related to lower stress and anxiety levels [8,25,26]. This could be one reason why most individuals kept exercising during the pandemic, as indicated in the present study and confirmed by the current literature [27]. Additionally, physical activity has shown a positive impact on dietary choices and mood states [28]. However, during this study, physical activity did not counterbalance the increased caloric intake, possibly due to emotional eating during the lockdown. Participants reported an increase in the consumption of high-calorie comfort foods, likely driven by emotional eating due to heightened stress and anxiety. Although there was a noted increase in home-based physical activities, these were insufficient to offset the caloric surplus. This discrepancy can be attributed to the lower intensity and shorter duration of home workouts, which are less effective in counteracting the effects of increased caloric intake. The constraints of home environments, such as limited space and lack of equipment, likely resulted in less vigorous physical activity than needed to effectively manage weight gain during the lockdown. The psychological strain of the lockdown, including increased isolation and stress, exacerbated tendencies towards emotional eating, further contributing to weight gain. Participant responses indicated a significant shift towards eating for comfort, highlighting the emotional dimensions influencing dietary habits during this period. In summary, while physical activity generally supports weight management, during the lockdown, it did not adequately counterbalance the increased caloric intake associated with emotional eating. This underscores the need for holistic health strategies that address both physical activity and emotional well-being during prolonged periods of stress [29,30,31,32,33,34,35].

Collectively, all these changes in lifestyle habits due to the lockdown resulted in weight gain. No significant differences were found between males and females. This is interesting because another study’s results indicated that females during the COVID-19 lockdown were more preoccupied with food and experienced decreased body image satisfaction compared to males [29]. Hence, this sex-specific difference in body image perception may not be indicative of objective body image formation and should be treated with caution. Additionally, the results of weight gain due to potentially poor-quality dietary behavior could trigger restrictive nutritional traits in adults, which are shown to be associated with underaged individuals’ adiposity since early adulthood [30].

Our findings indicate that supermarkets were considered the safest places to purchase food during the COVID-19 lockdown, followed by online sales. This perception likely reflects the public’s trust in the hygiene measures and social distancing protocols implemented by supermarkets. The controlled environment of supermarkets, including regulated entry and exit points, frequent sanitization of surfaces, and the enforcement of mask-wearing, may have contributed to their perception as safe spaces [21,30]. The low percentage of considering the public market safe could be attributed to the challenges public markets face in enforcing social distancing and other preventive measures due to their typically open and crowded nature. Additionally, the variability in individual vendor practices regarding hygiene could contribute to the perception of increased risk. The distinction in safety perception between physical stores and online sales also highlights an important aspect of consumer behavior during the pandemic. While physical safety concerns dominate in supermarkets, online purchases are perceived through the lens of transaction security and the avoidance of physical contact, which were significant concerns during the peak of the COVID-19 crisis. These perceptions of safety have significant implications for public health policy and retail management, especially in preparing for future public health emergencies. Enhancing safety measures in public markets, improving communication about these measures, and possibly integrating more stringent standards for vendor operations could alter public perceptions and make these spaces safer for consumers. Furthermore, the growth in the preference for online shopping underscores the need to strengthen cybersecurity measures and ensure reliable delivery services to support public adherence to social distancing guidelines [23,30].

Another interesting finding was that employees in the health sector, who continued working during the COVID-19 lockdown, gained less weight compared to employees of other sectors. This is thought-provoking, as the literature suggests that in response to COVID-19-induced stress, people experienced higher levels of anxiety, fear, and anger [31], increasing the prevalence of emotional eating [32]. However, in this scenario, continuing to work appeared to compensate for this stress, resulting in less weight gain compared to other employees. Furthermore, those working in the health sector had less time for eating and engaged in more occupational physical activity than employees of other sectors [33].

In this manuscript, the reference to emotional eating is linked to the observed increase in the consumption of comfort foods during the COVID-19 lockdown, as reported in the results section. Specifically, the finding that a significant number of participants reported an increased intake of high-calorie, high-carbohydrate foods such as sweets and snacks can be indicative of emotional eating behaviors. This increase is discussed in the context of the psychological stress experienced by many during the lockdown, which often leads individuals to seek comfort in food as a coping mechanism. Comfort food consumption is a behavior commonly associated with emotional eating, where individuals eat not out of hunger but to satisfy emotional needs or to relieve stress [34,35]. The observed increase in the consumption of snacks and sweets during the lockdown may reflect emotional eating behaviors, which are commonly triggered by heightened stress, anxiety, or boredom. Emotional eating involves the consumption of high-calorie, energy-dense foods in response to negative emotional states rather than physiological hunger. This phenomenon has been well-documented in the literature as a coping mechanism during periods of uncertainty and isolation, such as the COVID-19 pandemic.

The significant lifestyle changes brought about by the lockdown, including altered daily routines, increased time spent at home, and reduced opportunities for social interaction, may have exacerbated stress levels, contributing to emotional eating behaviors. Additionally, the closure of gyms and restrictions on outdoor activities may have further limited alternative coping mechanisms, such as physical activity, which is often associated with stress relief [31,32].

The observed increase in caloric intake, particularly from snacks and comfort foods, coupled with changes in physical activity, suggests a potential long-term risk for weight gain and associated metabolic conditions, such as type 2 diabetes and cardiovascular disease, if these behaviors persist. Conversely, increased fruit and vegetable consumption among some participants may provide a protective effect. These findings highlight the importance of targeted interventions to encourage balanced nutrition and sustained physical activity during periods of restricted mobility to mitigate the risk of chronic disease development [36,37].

The finding that 74.8% of participants reported being non-smokers at the time of the COVID-19 study, while only 25.2% were smoking cigarettes, is noteworthy. This high prevalence of non-smoking behavior among the study population may reflect increased health awareness and behavior modification during the pandemic. The COVID-19 pandemic has been associated with heightened public awareness of respiratory health, which could have motivated individuals to reduce or quit smoking to lower their risk of severe illness. Additionally, the stress and anxiety associated with the pandemic might have had varying effects on smoking behavior. While some individuals may have used smoking as a coping mechanism, others might have taken the opportunity to adopt healthier lifestyles, including smoking cessation. Public health campaigns and smoking cessation programs could have also played a role in encouraging individuals to quit smoking during this period. The relatively low percentage of smokers (25.2%) in our study population suggests that targeted interventions to support smoking cessation could be beneficial, particularly in the context of ongoing public health efforts to mitigate the impact of COVID-19. Further research is needed to explore the factors influencing smoking behavior during the pandemic and to identify effective strategies for promoting long-term smoking cessation. Overall, these findings underscore the importance of continued public health initiatives aimed at reducing smoking prevalence and supporting individuals in maintaining smoke-free lifestyles, especially during times of heightened health risks such as the COVID-19 pandemic [22,37].

Shakoor et al. (2021) emphasize that both vitamin C and vitamin D play complementary roles in supporting the immune system, reducing the risk and severity of respiratory infections, and potentially aiding in the management of viral infections like COVID-19. These nutrients are essential components of immune health and may have a preventive and therapeutic role in mitigating the impact of infections [38].

### 4.1. Strengths and Limitations

The present study offers several notable strengths. Firstly, the large number of sample and the timeliness and relevance of the survey are significant. Conducted expeditiously during the critical initial phase of the COVID-19 pandemic and lockdown in Cyprus and Greece, this study captures real-time data on how individuals adapted their lifestyles in response to unprecedented conditions. Secondly, the study provides a comprehensive lifestyle assessment, offering insights into weight changes and lifestyle habits during the lockdown. This enhances our understanding of individual behavioral adaptations and weight development during the pandemic. Thirdly, the innovative data collection method is a strength. Although live interviews were not feasible due to COVID-19 restrictions, the online survey was designed to simulate face-to-face interactions, ensuring comprehensive data collection while adhering to safety guidelines.

Despite these strengths, this study also faces several limitations. The primary limitation is the reliance on self-reported data through a custom-made questionnaire. While the questionnaire was validated for consistency, self-reporting can lead to potential biases or inaccuracies in the data, such as underreporting or overreporting certain behaviors. Another significant drawback is the lack of objective measurements. The inability to conduct live interviews and objectively measure somatometric data and sleep-related changes is a notable limitation. Previous research indicates that sleep quality, which is closely linked to nutrition and exercise, was adversely affected during the lockdown. Objective measurements would have provided a more comprehensive analysis of these interdependencies.

Additionally, the sample size was not determined through statistical power calculations but was instead based on the maximum number of participants that could be recruited during the data collection period. This non-empirical approach, while practical, may affect the generalizability of the findings. Lastly, as an observational study, it was designed primarily to describe changes in eating, shopping habits, and lifestyle rather than to test specific hypotheses or estimate the effect size of interventions.

In summary, while the study offers valuable insights into lifestyle changes during the COVID-19 pandemic, it is important to consider these limitations when interpreting the findings.

### 4.2. Future Implications

Despite these limitations, the findings provide a valuable foundation for future research. They allow for comparisons with data from other countries and can inform the design of nutrition and lifestyle interventions tailored to circumstances of increased stress and anxiety. Future studies should consider incorporating objective measurements and structured psychological assessments to build on the insights gained from this research. While our study did not directly assess emotional eating or psychological states, the increased consumption of snacks and sweets aligns with findings from other studies that have reported similar patterns of stress-induced eating during the pandemic. Future research should incorporate validated psychological scales, such as the Emotional Eating Scale or Perceived Stress Scale, to better understand the relationship between emotional states and dietary changes observed during crises [39].

## 5. Conclusions

In summary, the COVID-19 lockdown had various impacts on the eating habits, food purchasing behaviors, and lifestyles of Cypriots and Greeks. One notable effect was an overall increase in participants’ weight with possible long-term implications for their health. Nutrition patterns changed, with participants increasing the frequency of daily meals and consuming higher amounts of fruits, vegetables, sweets, and snacks. Emotional eating was prevalent among many participants, even though exercise routines were not interrupted by the lockdown. Interestingly, employees in the health sector experienced less weight gain, likely due to their continued work during this period.

The finding that employees in the health sector experienced less weight gain suggests that continued work and potentially higher levels of physical activity during the lockdown might have mitigated the negative effects on weight. This finding emphasizes the importance of maintaining regular physical activity, even during challenging circumstances like a lockdown. If sustained, the dietary and lifestyle changes observed during the COVID-19 lockdown may increase the risk of chronic diseases, such as obesity and cardiovascular disease, emphasizing the need for proactive public health strategies to promote balanced nutrition and regular physical activity during similar events.

Overall, these findings highlight the need for targeted interventions and support to address the negative effects of lockdowns on individuals’ eating behaviors and weight management. Efforts should focus on promoting healthy dietary choices, managing emotional eating, and encouraging regular physical activity, particularly among populations experiencing increased stress and anxiety.

## Figures and Tables

**Figure 1 nutrients-17-00214-f001:**
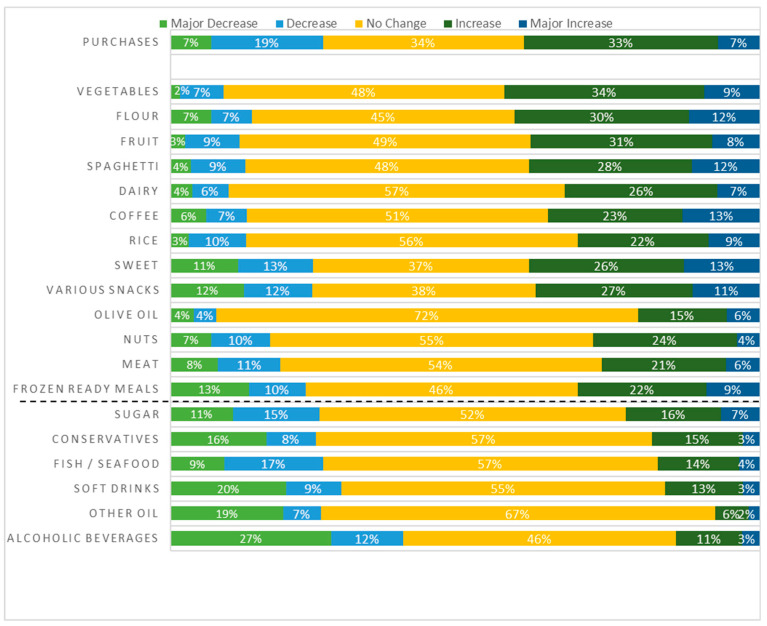
COVID-19 impact on food purchasing habits. The threshold was drawn on the level where the decrease percentage was higher than the increase.

**Table 1 nutrients-17-00214-t001:** Participants’ sociodemographic characteristics.

Variable	Category	n	%
Gender	Male	283	28
	Female	728	71.9
Age	17–25	409	40.5
	26–40	358	35.4
	41+	244	24.2
Country of origin	Cyprus	593	58.7
	Greece	388	38.4
	Other	30	2.9
Location	Rural	845	83.6
	Urban	141	13.9
	Other	25	2.5
Education	No degree	105	10.4
	BA	518	51.2
	MA+	388	38.4
Employment	Employed	655	64.8
	Unemployed	49	4.8
	Home Employment	15	1.7
	Retired	13	1.3
	Student	277	27.4
Profession	Health Professional	216	21.4
	Management Support	139	13.7
	Student	258	25.5
	Other	398	39.4
Marital Status	Single/Divorced/Widow	511	50.5
	Married/Cohabitant	500	49.5
Household Members	1	116	11.5
	2	241	23.8
	3	236	23.3
	4	280	27.7
	5+	139	13.7
Net monthly Income	No Income	199	19.7
	Less than 1000	303	30.0
	1001–2000	296	29.3
	2001–3000	127	12.6
	More than 3000	42	4.2
	No Answer	42	4.2
BMI (kg/m^2^)	<18.4	51	5.0
	18.5–24.9	631	62.4
	25.0–29.9	246	24.3
	>30.0	83	8.3

**Table 2 nutrients-17-00214-t002:** Estimated BMI distribution by gender, age, and country with percentages *.

BMI (kg/m^2^)	Total (n = 1011)	Male (n = 283)	Female (n = 727)	Age 17–25 (n = 409)	Age 26–40 (n = 358)	Age 41+ (n = 244)	Cyprus (n = 593)	Greece (n = 388)
<18.4	51 (5.0%)	14 (4.9%)	37 (5.1%)	20 (4.9%)	16 (4.5%)	15 (6.1%)	32 (5.4%)	19 (4.9%)
18.5–24.9	631 (62.4%)	177 (62.5%)	454 (62.5%)	255 (62.3%)	223 (62.3%)	153 (62.7%)	370 (62.4%)	261 (67.3%)
25.0–29.9	246 (24.3%)	69 (24.4%)	177 (24.3%)	100 (24.4%)	87 (24.3%)	59 (24.2%)	145 (24.5%)	101 (26.0%)
>30.0	83 (8.3%)	23 (8.1%)	60 (8.3%)	34 (8.3%)	32 (8.9%)	17 (7.0%)	46 (7.8%)	37 (9.5%)

* Numbers were rounded to the nearest whole number.

**Table 3 nutrients-17-00214-t003:** Meal frequency pre- and during COVID-19 lockdown.

Meal	Pre	During	*p*-Value	Difference
Breakfast	782 (77.3%)	824 (81.5%)	0.022	3.8% *
Midmorning Snack	428 (42.3%)	481 (47.6%)	0.353	−1.8%
Lunch	948 (93.8%)	949 (93.9%)	0.341	−1.1%
Afternoon	478 (47.3%)	626 (61.9%)	<0.001	12.2% **
Dinner	918 (90.8%)	938 (92.8%)	1.000	0.0%
Before Sleep	166 (16.4%)	298 (29.5%)	<0.001	18.5% **

** *p*< 0.01, * *p* < 0.05; McNemar’s test.

**Table 4 nutrients-17-00214-t004:** Daily food group consumption per serving, before and during COVID-19 lockdown.

Food Category ^a^	Serving Before Lockdown ^b^	Servings During Lockdown ^b^	z-Score	Significance
Various Snack	1	2	8.376	**
Sweet	2	2	7.154	**
Fruit	2	2	6.926	**
Vegetables	2	2	6.417	**
Legumes	2	2	3.625	**
Spaghetti/Rice	2	2	3.388	**
Soft drink	1	1	3.204	**
Meat	2	2	2.030	*
Fish/seafood	2	2	1.617	
Coffee	2	2	1.474	
Alcohol	1	1	−2.103	*

^a^ Food Category: Lists the types of food evaluated; ^b^ Servings: Median number of servings of each food category consumed by participants before and during the lockdown, respectively; * *p* < 0.05, ** *p* < 0.01; the z-scores are derived from the Wilcoxon signed-rank test, comparing median servings before and during the lockdown.

**Table 5 nutrients-17-00214-t005:** Daily food group frequency consumption, before and during COVID-19 lockdown.

Food Category ^a^	Frequency Before Lockdown ^b^	Frequency During Lockdown ^b^	z-Score	Significance
Various Snack	2	2	6.322	**
Sweet	2	2	5.385	**
Fruit	2	2	4.063	**
Vegetables	2	2	2.286	*
Legumes	1	1	1.356	
Spaghetti/Rice	3	3	0.909	
Soft drink	3	3	0.056	
Meat	3	3	-0.991	
Fish/seafood	2	2	-0.211	
Coffee	2	2	-1.364	
Alcohol	1	1	-2.918	**

^a^ Food Category: Lists the types of food evaluated; ^b^ Median frequency of consumption of each food category before the and during the lockdown, respective; * *p* < 0.05, ** *p* < 0.01; the z-scores are derived from the Wilcoxon signed-rank test, comparing median frequencies before and during the lockdown.

**Table 6 nutrients-17-00214-t006:** Exercise habits, before and during COVID-19 lockdown.

Physical Activity	Frequency Before (Median)	Frequency During (Median)	z-Score ^a^	Significance
Home Gymnastics	1	2	15.286	**
Running/Walking	2	2	2.918	**
Cycling	1	1	1.294	
Dance	1	1	−0.608	
Yoga/Pilates	1	1	−0.847	
HIIT	1	1	−1.383	
TRX	1	1	−5.428	**
Swimming	1	1	−5.540	**
Weightlifting	1	1	−6.529	**
Gym	1	1	−13.820	**

^a^ The z-scores are derived from the Wilcoxon signed-rank test, comparing median frequencies before and during the lockdown. ** *p* < 0.01.

**Table 7 nutrients-17-00214-t007:** Nutritional supplement usage and perceptions during COVID-19 pandemic.

Description	Percentage (%)	Chi-Square/X^2^ (if Applicable) *
Participants taking supplements	39%	
Multivitamins	11.5%	
Vitamin D	16.2%	
Vitamin C	13.9%	
Belief in supplements helping against COVID-19	34%	
No opinion on supplements	37%	
Believe supplements do not help	30%	
Comparison of beliefs (supplement users vs. non-users)		X^2^ (2, N = 1011) = 63.163, *p* < 0.001

* *p* < 0.001; The chi-square (X²) test was used to compare beliefs regarding the effectiveness of supplements against COVID-19 between supplement users and non-users.

**Table 8 nutrients-17-00214-t008:** Summary of weight and BMI changes during COVID-19 lockdown.

Metric	Before Lockdown Mean (SD)	During-Lockdown Mean (SD)	t-Value	*p*-Value	Change
Weight (kg)	67.584 (14.978)	68.04 (15.359)	3.988	<0.001 *	+0.456 kg
BMI (kg/m²)	23.8 (4.35)	23.98 (4.201)	3.176	<0.001 *	+0.18 kg/m²

* *p* < 0.001; paired sample *t* test.

**Table 9 nutrients-17-00214-t009:** Distribution of weight changes among participants.

Weight Change Category	N (%)	Mean Change (kg)	Range of Change (kg)
Weight Gain	425 (41)	+2.34	0.5 to 17
Weight Maintenance	425 (41)	0	0
Weight Loss	182 (18)	−2.2	1 to 6

**Table 10 nutrients-17-00214-t010:** Demographic influences on weight changes.

Demographic Factor	Comparison	t-Value *	*p*-Value *	Correlation (r) **
Gender	Male vs. Female	0.667	0.505	N/A
Location	Cyprus vs. Greece	−1.747	0.081	N/A
Age			0.151	0.05
Occupation (Health Sector vs. Others)		−1.980	0.048	N/A

* The t-values and *p*-values are derived from *t*-tests comparing before and during lockdown measures. ** The correlation (r) value is from Pearson correlation analysis. N/A: Not Applicable.

## Data Availability

The data presented in this study are available on request from the corresponding author.

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
