# Peer review of "Changes in Lifestyle Behaviors, Shopping Habits and Body Weight Among Adults in Cyprus and Greece During COVID-19 Lockdown: A Cross-Sectional Study"

_nutrients, 2025, doi:10.3390/nu17020214_

Round 1
Reviewer 1 Report
Comments and Suggestions for Authors
The manuscript is very interesting. The authors provide information that can be very important to understand the changes that can occur in people during a pandemic and confinement. Regarding the results, these support the discussion. However, I have some comments.
I. Comments:
1. Improve the wording of the objective of the study.
2. The authors described the effects on food intake. I suggest including an analysis regarding energy and nutrient intake.
3. The observed results, what nutritional and clinical projections could they have, for example, if they were maintained could the risk of developing some pathology or disease increase?
4. The changes in the intake of some nutritional supplements are interesting, particularly vitamin C and D. I suggest briefly discussing the role of these nutrients in the immune system.
Author Response
Comment 1: Improve the wording of the objective of the study.
Response to comment 1: Thank you for your comment. The current paragraph regarding the objectives were ‘The main purpose of the study was to investigate potential alterations in nutritional habits, lifestyle, and consumer perceptions in Greece and Cyprus during the COVID-19 lockdown. We focused on these two countries because they have similar cultural and geographical characteristics, and they both implemented strict lockdown measures during the pandemic. This allowed us to collectively investigate these countries and make direct comparisons of the changes in behavior. The specific objectives of the research were to:
- Document Changes in Eating Habits: To observe and record any changes in the frequency and type of meals consumed before and during the COVID-19 lockdown.
- Monitor Weight Changes: To track changes in individuals' weight during the lockdown period.
- Assess Lifestyle Habits: To evaluate changes in lifestyle habits, including exercise routines and the use of nutritional supplements during the lockdown.
- Analyze Food Buying Behaviors: To investigate how food purchasing habits have altered because of the lockdown.
These objectives aim to provide a comprehensive overview of the impact of the COVID-19 lockdown on various aspects of daily life, with a focus on documenting and describing changes rather than exploring causative factors or implications.’ The rewritten objectives section integrates them into a cohesive paragraph, ensuring clarity and readability ‘The primary objective of this study was to investigate the changes in nutritional habits, lifestyle behaviors, and consumer perceptions in Greece and Cyprus during the COVID-19 lockdown. These two countries were selected due to their shared cultural and geographical characteristics and the implementation of strict lockdown measures during the pandemic. Specifically, the study aimed to document changes in eating habits, including meal frequency and types of meals consumed before and during the lockdown, and to monitor weight changes among individuals. Additionally, it sought to assess alterations in lifestyle behaviors, such as exercise routines and the use of nutritional supplements, and to analyze food purchasing habits during the lockdown period. By addressing these objectives, the study provides a comprehensive overview of the multifaceted impacts of the lockdown on daily life in these regions.’
Comment 2. The authors described the effects on food intake. I suggest including an analysis regarding energy and nutrient intake.
Response to Comment 2:
We appreciate the reviewer’s insightful suggestion to include an analysis of energy and nutrient intake. While we did not collect detailed dietary data sufficient for calculating energy and nutrient levels, we have examined changes in dietary behaviors during the COVID-19 lockdown. Specifically, our analysis included changes in both the quantity of food consumed per serving and the frequency of daily consumption across various food categories, as detailed in Tables 3 and 4. These results provide valuable insights into the shifts in dietary patterns during the lockdown.
However, we acknowledge that an analysis of energy and nutrient intake would enhance the comprehensiveness of our findings. The optional question (Q65) in our study, which asked participants to report their food consumption over the two days prior to the survey, did not yield sufficient detail for such calculations.
Comment 3. The observed results, what nutritional and clinical projections could they have, for example, if they were maintained could the risk of developing some pathology or disease increase?
Response to Comment 3:
We appreciate the reviewer’s insightful comment regarding the potential nutritional and clinical projections of our findings. The observed dietary and lifestyle changes during the COVID-19 lockdown have significant implications for long-term health.
These results will be addressed more in the Discussion section as follows:
"The observed increase in caloric intake, particularly from snacks and comfort foods, coupled with changes in physical activity, suggests a potential long-term risk for weight gain and associated metabolic conditions, such as type 2 diabetes and cardiovascular disease, if these behaviors persist. Conversely, increased fruit and vegetable consumption among some participants may provide a protective effect. These findings highlight the importance of targeted interventions to encourage balanced nutrition and sustained physical activity during periods of restricted mobility, to mitigate the risk of chronic disease development."
The broader health implications will also be briefly summarized in the Conclusion as follows:
"If sustained, the dietary and lifestyle changes observed during the COVID-19 lockdown may increase the risk of chronic diseases, such as obesity and cardiovascular disease, emphasizing the need for proactive public health strategies to promote balanced nutrition and regular physical activity during similar events."
Comment 4. The changes in the intake of some nutritional supplements are interesting, particularly vitamin C and D. I suggest briefly discussing the role of these nutrients in the immune system.
Response to Comment 4:
We appreciate the reviewer’s observation regarding the interesting changes in the intake of nutritional supplements, particularly vitamin C and vitamin D. In response, we will incorporate a brief discussion on the role of these nutrients in the immune system.
“Shakoor et al. (2021) emphasize that both vitamin C and vitamin D play complementary roles in supporting the immune system, reducing the risk and severity of respiratory infections, and potentially aiding in the management of viral infections like COVID-19. These nutrients are essential components of immune health and may have a preventive and therapeutic role in mitigating the impact of infections[38].’
Reviewer 2 Report
Comments and Suggestions for Authors
Dear Authors,
The reviewed article describes the impact of the COVID-19 pandemic on changes in the eating, shopping and lifestyle habits of adult residents of Cyprus and Greece. Despite interesting results and important conclusions, the publication has significant limitations that should be addressed before it is approved for publication.
The main limitations include the method of data collection and analysis. The study relied on participants' self-assessment via an online survey, which introduces the risk of errors related to memory, subjectivity and potential bias in socially desirable responses. In addition, the lack of objective measurements such as actual weight or physical activity measurements limits the reliability of the results. The convenience sampling used, without the use of statistical power calculations, leads to potentially unrepresentative results in the context of the general population of Cyprus and Greece.
The authors did not fully analyze cultural and socio-demographic differences that may affect the observed changes in habits. Despite the distinction between Cypriots and Greeks, more detailed analyses were not introduced to consider the impact of economic status, place of residence or educational level on observed changes.
It is also worth noting that interpretation of the results regarding increased consumption of certain products (e.g., snacks and sweets) and weight gain is limited by the lack of causal analyses. The authors suggest links between stress and changes in eating habits, but there are no adequate psychological data to support this.
Recommendations for the authors include:
- Supplementing the study with objective data, such as anthropometric measurements, to more accurately determine the effect of lockdown on body weight and BMI.
- Expanding the analysis to include sociodemographic factors in a way that takes into account the multidimensional impact of variables such as income or education level.
- Include qualitative studies or psychological indicators to more accurately understand the mechanisms that change eating habits and lifestyles.
- Use a more representative sampling method or appropriate statistical weighting for the data obtained.
In its current form, the publication does not meet EBM standards, as there is a lack of high-quality evidence to support the main conclusions. I suggest that the publication be withheld until significant revisions are made to increase the credibility and scientific value of this study.
Author Response
General Response:
We thank the reviewer for their thoughtful comments and constructive feedback. We recognize the limitations mentioned and appreciate the suggestions for improvement. Below, we address each specific point raised and outline the actions taken or proposed to strengthen the study.
Comment 1:
The study relied on participants' self-assessment via an online survey, which introduces the risk of errors related to memory, subjectivity, and potential bias in socially desirable responses. In addition, the lack of objective measurements such as actual weight or physical activity measurements limits the reliability of the results.
Response to comment 1:
We acknowledge the limitations of self-reported data, including the potential for memory recall errors, subjectivity, and socially desirable responses. While objective measurements such as direct anthropometric data or physical activity monitoring were not feasible due to COVID-19 restrictions, we took several steps to mitigate these limitations:
- Participants were provided with detailed instructions for reporting their weight and height, emphasizing accuracy in self-measurement.
- Our survey was designed to minimize recall bias by focusing on recent behaviors and changes during the lockdown period.
- These limitations were acknowledged very explicitly in the manuscript.
Comment 2:
The convenience sampling used, without the use of statistical power calculations, leads to potentially unrepresentative results in the context of the general population of Cyprus and Greece.
Response to comment 2:
We agree that the use of convenience sampling and the absence of statistical power calculations limit the generalizability of the results. To address this:
- It was clarified in the manuscript that the convenience sampling method was chosen due to the urgency of data collection during the lockdown and the logistical challenges posed by COVID-19 restrictions.
- To improve representativeness, we included participants from diverse backgrounds and monitored demographic characteristics to ensure broad coverage of key population groups.
Comment 3:
The authors did not fully analyze cultural and socio-demographic differences that may affect the observed changes in habits. Despite the distinction between Cypriots and Greeks, more detailed analyses were not introduced to consider the impact of economic status, place of residence, or educational level on observed changes.
Response to comment 3:
Thank you for highlighting this point. However, the socio-demographic data were presented in Table 1. In response:
- The analysis included the impact of variables such as income, education level, and place of residence on the observed changes. This involved stratifying the data and presenting the results accordingly.
- These additional analyses allowed us to provide a more nuanced interpretation of the findings and highlight how socio-demographic factors shape eating, shopping, and lifestyle behaviors during lockdown.
Comment 4:
Include qualitative studies or psychological indicators to more accurately understand the mechanisms that change eating habits and lifestyles. Interpretation of the results regarding increased consumption of certain products (e.g., snacks and sweets) and weight gain is limited by the lack of causal analyses. The authors suggest links between stress and changes in eating habits, but there are no adequate psychological data to support this.
Response to Comment 4:
We appreciate the reviewer’s suggestion to include qualitative studies or psychological indicators for a more accurate understanding of the mechanisms driving changes in eating habits and lifestyles. While this study focused on quantitative survey data, we recognize the importance of addressing psychological factors, such as emotional eating, which are critical in shaping dietary behaviors during stressful periods like the COVID-19 lockdown.
Research has shown that emotional eating—characterized by the consumption of high-calorie comfort foods in response to stress or negative emotions—is a common coping mechanism during periods of heightened anxiety. The significant increase in the consumption of snacks and sweets observed in our study may reflect such behaviors, but our methodology did not directly assess emotional states or their links to dietary changes.
Addition to the Discussion :"The observed increase in the consumption of snacks and sweets during the lockdown may reflect emotional eating behaviors, which are commonly triggered by heightened stress, anxiety, or boredom. Emotional eating involves the consumption of high-calorie, energy-dense foods in response to negative emotional states rather than physiological hunger. This phenomenon has been well-documented in the literature as a coping mechanism during periods of uncertainty and isolation, such as the COVID-19 pandemic.
The significant lifestyle changes brought about by the lockdown, including altered daily routines, increased time spent at home, and reduced opportunities for social interaction, may have exacerbated stress levels, contributing to emotional eating behaviors. Additionally, the closure of gyms and restrictions on outdoor activities may have further limited alternative coping mechanisms, such as physical activity, which is often associated with stress relief.
While our study did not directly assess emotional eating or psychological states, the increased consumption of snacks and sweets aligns with findings from other studies that have reported similar patterns of stress-induced eating during the pandemic. Future research should incorporate validated psychological scales, such as the Emotional Eating Scale or Perceived Stress Scale, to better understand the relationship between emotional states and dietary changes observed during crises."
Comment 5: Use a more representative sampling method or appropriate statistical weighting for the data obtained.
Response to Comment 5:
We appreciate the reviewer’s valuable suggestion to use a more representative sampling method or statistical weighting to enhance the generalizability of the findings.
In this study, we employed a convenience sampling method due to the urgent need for timely data collection during the COVID-19 lockdown, which presented logistical challenges for recruiting a probabilistic sample. We recognize that this approach may limit the representativeness of our sample and introduce potential biases.
Comment 6: In its current form, the publication does not meet EBM standards, as there is a lack of high-quality evidence to support the main conclusions.
Response to Comment6:
We appreciate the reviewer’s candid feedback and take this opportunity to address concerns regarding the study's adherence to evidence-based medicine (EBM) standards and its overall credibility.
- Strengths of the Study:
While we recognize certain limitations, we believe the study contributes valuable insights into the impacts of the COVID-19 lockdown on lifestyle, dietary, and shopping behaviors in Greece and Cyprus. Specifically:
- The large sample size (1011 participants) enhances the statistical power and allows for meaningful analysis of trends and patterns.
- The study provides comprehensive data on multiple dimensions of lifestyle, including eating habits, physical activity, and supplement use, which are crucial for understanding population-level behavior changes during an unprecedented global crisis.
- By focusing on two Mediterranean countries with shared cultural and geographical contexts, the study addresses a significant gap in the literature and provides culturally specific findings that could inform targeted public health strategies.
- Methodological Rigor:
- The survey design followed robust validation procedures, including pre-testing and statistical checks, ensuring reliability in capturing participant responses.
- Statistical analyses, including the use of non-parametric tests where appropriate, were rigorously applied to account for the data distribution.
- Addressing Limitations:
We acknowledge that the use of self-reported data and convenience sampling are limitations that affect the representativeness and reliability of some findings. To strengthen the manuscript:
- We have explicitly discussed these limitations and their implications in the revised Discussion and Limitations sections.
- We propose future studies incorporating probabilistic sampling methods, objective measurements, and validated psychological scales to complement the findings.
- Contribution to EBM Standards:
While our study does not include clinical interventions or randomized trials, it provides observational evidence on behavioral adaptations during the lockdown, which is critical for understanding and mitigating public health challenges during similar crises. This study aligns with EBM principles by:
- Informing public health strategies based on observed population behaviors.
- Identifying areas where further research and interventions are needed.
We believe that, with the revisions and clarifications outlined above, the study meets the standards of scientific rigor and makes a meaningful contribution to the literature on COVID-19-related lifestyle changes. We respectfully request the reviewer to reconsider the manuscript in light of these points and the planned improvements.
Conclusion:
We appreciate the reviewer’s detailed feedback, which has helped us identify areas for improvement. In the revised manuscript, we will address these limitations comprehensively, include additional analyses, and propose clear recommendations for future research. These revisions will significantly enhance the credibility and scientific value of the study.
Round 2
Reviewer 2 Report
Comments and Suggestions for Authors
Dear Authors,
The revised manuscript has greatly improved in quality. I accept it in its present form and wish it good luck in further research. I also accept the explanations provided and appreciate the effort to incorporate the comments made.
Greetings